# Exploring the impact of narrowing urban-rural income gap on carbon emission reduction and pollution control

**Lujing Wang**[1,2], **Ming Zhang**[1,2]*

**1** School of Economics and Management, China University of Mining and Technology, Xuzhou, China,
**2** Center for Environmental Management and Economics Policy Research, China University of Mining and Technology, Xuzhou, China

* zhangmingdlut@163.com

**Data Availability Statement:** All relevant data are in the Supporting Information files.

**Funding:** The author(s) received no specific funding for this work.

## Abstract

Over the past four decades, China have experienced rapid economic growth but also a widening urban-rural income gap and deteriorating air quality. Based on the panel data of 30 provinces in China from 2006 to 2017, this paper investigates the effect of narrowing the urban-rural income gap on carbon emission reduction and pollution control by using OLS method. The empirical results indicate that: the narrowing of the urban-rural income gap has a positive impact on pollution control, while there are regional differences in the impact on carbon emission reduction. In the perspective of the whole country and central and western regions, the narrowing of the urban-rural income gap is conducive to carbon emission reduction. However, the narrowing of the urban-rural income gap increases carbon emissions in the eastern regions where economic development is at high level. This paper provides a theoretical basis and policy reference for promoting urban-rural integration and construction of ecological civilization.

## Introduction

The widening urban-rural income gap and the deterioration of ecological environment are two major problems in the process of China's urbanization and industrialization [1, 2]. For a long time, the urban-rural income gap has been at a high level in China. Differences in education and health care levels, imperfect social security systems and a tax system based on regressive taxes further exaggerate the urban-rural income gap [3]. The widening urban-rural income gap not only destroys the benign competitive atmosphere of the society, but also causes social crisis and turmoil when a certain limit is reached. Meanwhile, extensive economic development has also tightened resource constraints and overloaded the ecosystem's ability to purify itself in China. According to the China Ecological Environment Status Bulletin in 2019, more than half of the country's cities do not meet air quality standards. With the improvement of living standards, residents have put forward higher level requirements for ecological environment quality. However, the difference in economic basis between urban and rural residents determines the demand preference for environmental quality which in turn has an impact on

**Competing interests:** The authors have declared that no competing interests exist.

the environment [4]. Therefore, clarifying the relationship between urban-rural income gap and environmental pollution is conducive to grasping the evolutionary rules in the process of new urbanization and conducting targeted social governance.

The Kuznets Curve (KC) proposed by Kuznets suggested that the income gap will widen and then narrow in a developing country or region. There is an inverted "U" nonlinear relationship between economic development and the income gap [5]. On this basis, Grossman and Krueger concluded that increasing income causes changes in the environmental quality of a country or region that deteriorate and then improve. This is the Environmental Kuznets Curve (EKC). Based on the above theories, we argues that there must be a certain connection between income distribution and environmental pollution.

Existing studies on income gap and environmental pollution can be broadly divided into two types of literature. The first one is based on the negative external hypothesis of environmental pollution. It uses healthy human capital as a mediator, to draw general conclusions that environmental pollution led to the widening of the income gap [6]. Therefore, the issue of this study is proposed that environmental pollution has an impact on income distribution by affecting health human capital. Environmental pollution is an important cause of damage to physical and mental health. Living in a poor environment will not only cause respiratory infections and other air pollution-related diseases, but also induce various chronic mental illnesses [7–10]. Health, as an indispensable human capital for human development, is a necessary factor to ensure productivity. Environmental pollution increases human health risks and reduces labor supply, which in turn has a negative impact on productivity [11, 12]. As environmental pollution becomes more severe, the gap between the labor capital held by the two becomes more pronounced. By constructing an intertemporal economic model, Sheng found that low-income rural residents invest less human capital in health damage caused by environmental pollution [13]. Thus, with the gradual widening of the health gap between urban and rural residents, the income gap between urban and rural residents may increase as the aggravation of pollution.

The second one is the study about the impact of income gap on environmental pollution. The related literature suggested that higher income gap can have an impact on environmental pollution through the behavioral choices of people. Boyce was the first to analyze the behavior of different income groups towards environmental pollution [14]. He argued that inequalities in power and income will exacerbate environmental pollution. Ala-Mantila et al. considered that previous studies had only considered the propensity to consume and ignored the environmental impact of saving behavior [15]. Even though high-income people had a low marginal propensity to consume and a high marginal amount of savings motivates productive investment, which can also put pressure on the environment. Income gap not only affected individual consumption and saving behavior, but also had a negative impact on business innovation. Vona and Patriarca developed an asymmetric model of environmentally friendly and non-environmentally friendly products to demonstrate that unequal income distribution had a significant negative impact on environmental innovation from the perspective of environmental innovation [16]. The widening income gap severely hinders the diffusion of environmentally friendly technologies and reduces their positive externalities. Based on an endogenous growth model, Eriksson and Persson argued that poor middle voters have weaker preferences for environmental quality, and the lower the income, the lower the marginal rate of substitution between environmental consumption and private consumption [17]. It would lead individuals to make decisions with a preference for the consumption of private goods over the environment, and conversely, the richer the middle voter, the higher the environmental tax rate and the better the environmental quality. Baek and Gweisah, Kasuga and Takaya, Kusumawardani and Dewi found that widening income disparity exacerbates environmental pollution through

empirical analysis [18–20]. Scruggs, Herrink, Hubler differed from the view above [21–23]. They assumed that the environment is a luxury and the rich have higher environmental qualities and can afford to consume environmentally friendly products, which the widening income gap may not bring about a worsening of environmental pollution or even benefit ecological protection. However, the above studies did not consider the relationship between income distribution and environmental pollution under the effect of regional differences in economic development. By extending the framework of the environmental Kuznets curve, Chen found that equal income distribution in developing countries is beneficial in reducing carbon emissions, while in most developed countries, unequal income distribution has almost no impact on carbon emissions [24]. Zhang and Zhao showed that the reduction in income gap has a positive effect on the reduction of carbon emissions in both eastern and western China, but its effect is smaller in the east than in the west [25]. Thus, the existing literature on the relationship between income inequality and environmental pollution are still debated.

Existing studies do not sufficiently consider the differences in the relationship between different income groups on environmental issues. Higher-income groups tend to enjoy the benefits of economic growth, while lower-income groups bear more responsibility for the consequences of environmental pollution. China is a typical dual economy combining urban and rural areas. There are significant differences between urban and rural areas in terms of consumption patterns, industrial structures, and technology levels. Therefore, it is necessary to conduct an in-depth analysis of the relationship between urban-rural income disparity and environmental pollution in China. Based on the research of domestic and foreign scholars, this paper uses panel data for 30 Chinese provinces from 2006–2017 and empirically analyzes the impact of urban-rural income disparity on emission reduction and pollution control through the OLS method. Specifically, the marginal contributions of this paper are mainly the following four points: (1) In the selection of pollutants, both source emission reduction and end-of-pipe pollution control are considered to supplement and improve the existing studies. At the input side, most industries' production activities are dominated by the consumption of fossil energy and carbon emissions are higher. Therefore, carbon emissions are selected as the variable for carbon emission control at the production source. At the output side, the entropy weight method is used to construct the pollution control capacity index as the variable of pollution control at the output side based on the data of industrial three waste emissions. (2) Considering the possible two-way causality between urban-rural income gap and environmental pollution, this paper uses Granger non-causality test for ranking. And the lagged first order of urban-rural income gap is used as an instrumental variable to alleviate the endogenous problems in the model. (3) The relationship between urban-rural income disparity and emission reduction and pollution control was analyzed heterogeneously about the eastern, central and western regions in China.

## Model construction and data description

### Model construction

From the perspective of the production side and the output side, we selects per capita carbon emissions and end-of-pipe pollution control as the explanatory variables to verify the impact of urban-rural income gap on carbon emission reduction and pollution control, and the Theil Index and urban-rural income ratio as the proxy variables of urban-rural income gap. In order to verify the EKC curve, per capita GDP is used to measure the level of economic growth. At the same time, the quadratic and cubic terms of per capita GDP are introduced into the regression equation. In addition, traditional factors such as energy consumption per unit of GDP, environmental regulation intensity, urbanization rate, and industrial structure are introduced

into the equation as control variables. The following Eqs (1) and (2) are constructed as an econometric model based on panel data.

$$\ln EQ_{it} = \beta_0 + \beta_1 \ln GAP_{it} + \beta_2 \ln GDP_{it} + \beta_3 (\ln GDP_{it})^2 + \beta_4 (\ln GDP_{it})^3$$
$$+ \beta_5 \ln ERI_{it} + \beta_6 \ln ISO_{it} + \beta_7 \ln UR_{it} + \beta_7 \ln EE_{it} + \mu_{it}$$

(1)

$$\ln PerCO2_{it} = \partial_0 + \partial_1 \ln GAP_{it} + \partial_2 \ln GDP_{it} + \partial_3 (\ln GDP_{it})^2 + \partial_4 (\ln GDP_{it})^3$$
$$+ \partial_5 \ln ERI_{it} + \partial_6 \ln ISO_{it} + \partial_7 \ln UR_{it} + \partial_7 \ln EE_{it} + \varepsilon_{it}$$

(2)

## Selection of variables and data sources

In this paper, environmental pollution is measured in two dimensions of the production process. Firstly, in the context of global warming, various countries are actively responding to the call for energy saving and emission reduction. China also advocates the green development model. Therefore, carbon emissions per capita is selected as a proxy variable for "carbon emission reduction". Secondly, we use the entropy weight method to construct the end-of-pipe treatment intensity index as the proxy variable of "pollution control" by selecting the data of industrial wastewater emissions, industrial sulfur dioxide emissions and industrial solid emissions. In addition, using the urban-rural income ratio and the Theil index as proxy variables of the urban-rural income gap.

(1) Emission source reduction ($PerCO_2$). The increase in carbon emissions is mainly due to the combustion and use of fossil fuels. This paper draws on the calculation method of Du and divides fossil energy consumption into coal, coke, petroleum (gasoline, kerosene, vegetable oil, fuel oil), natural gas consumption and cement production emissions [26]. According to the CO2 emission coefficients of various emission sources, the total carbon emission of each province is finally obtained. The per capita carbon emission is calculated as a proxy variable for source emission reduction based on the total population (data from the National Statistical Yearbook, China Energy Statistical Yearbook and the provincial statistical yearbooks 2007–2018).

(2) End-of-pipe pollution control ($EQ$). In this paper, we adopt the global entropy weighting method based on Yuan, which is to expand the three-dimensional time series data table into a two-dimensional table from top to bottom in chronological order, and use the traditional entropy weighting method to assign weights to the dimensionless data. Industrial waste is considered as a very small indicator [27].The result is the end-point pollution control capacity of each province (data from National Statistical Yearbook and provincial statistical yearbooks 2007–2018).

(3) Urban-rural income ratio ($GAP$). Fig 1 shows the change in income of urban and rural residents since 1978. The reform and opening up introduced the market regulation under the original planned economic system. In 1992, China formally established the reform goal of socialist market economy. The non-public economy gradually grew, and the income as well as the living standard of urban and rural residents were greatly improved. In 2001, China introduced foreign investment and established a modernized enterprise system, which released the vitality of our market and further raised the income level of residents. Along with the increase of income level, the income gap between urban and rural areas is also expanding. The ratio of the two incomes reached its highest value in 2009. Although there has been a downward trend since then, the income of urban residents is still much higher than that of rural residents. The paper uses the ratio of urban residents' disposable income to rural residents' net income per capita (which becomes the disposable income of rural residents after 13 years) as an indicator

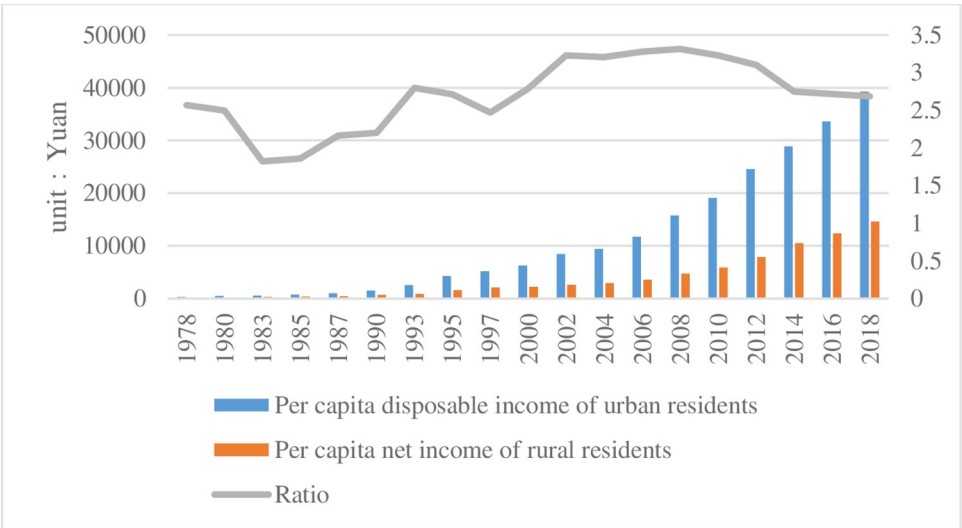

**Fig 1. Change in income of urban and rural residents since the reform and opening-up of China.**

of inequality in urban-rural income distribution (data from the National Statistical Yearbook and the provincial statistical yearbooks 2007–2018).

(4) The Theil index (*Theil*). The Theil index is an index that uses the concept of entropy in information theory to measure the income distribution gap between individuals or regions. It measures the intra-group gap and the contribution of the intra-group gap to the overall gap, and is sensitive to changes in high income levels. Drawing on Wang et al. [28], the Thiel index is calculated as

$$\text{Theil} = \sum_{i=1}^{2} \left[ \frac{Y_{ij,t}}{Y_{j,t}} \right] \ln \left[ \frac{Y_{ij,t}}{Y_{j,t}} \Big/ \frac{Z_{ij,t}}{Z_{j,t}} \right] \tag{3}$$

where $i = 12$ denote urban and rural areas, respectively. $Z_{i,t}$ denotes the urban or rural population in year $t$ in region $j$. $Y_{i,t}$ denotes total disposable income, and $Y_{ij,t}$ denotes disposable income of urban or rural residents.

Fig 2 shows the direction of changes in the Urban-Rural Thiel Index in each region from 2005 to 2018. The Urban-Rural Thiel Index shows a downward trend in nationally and the three regions of east, west and central, indicating that income distribution tends to be equal. From a regional perspective, the eastern region is not only ahead of the central and western regions in terms of economic growth, transportation and urbanization rate, but also has a relatively reasonable income distribution. The Urban-Rural Thiel index lower than the national level. The central Region remains largely consistent with the nation. There is a big income gap between urban and rural residents in the western region. Therefore, the east and west regions exist not only to coordinate inter-regional economic development, but also to develop development strategies that are in line with their own economic and social status in order to reduce the differences between regions and rationalize income distribution (Data from National Statistical Yearbook, China Population and Employment Statistical Yearbook and provincial statistical yearbooks 2007–2018).

In addition to the core variables mentioned above, some scholars found that other variables can also directly affect environmental pollution [29, 30], so we choose control variables as shown in Table 1.

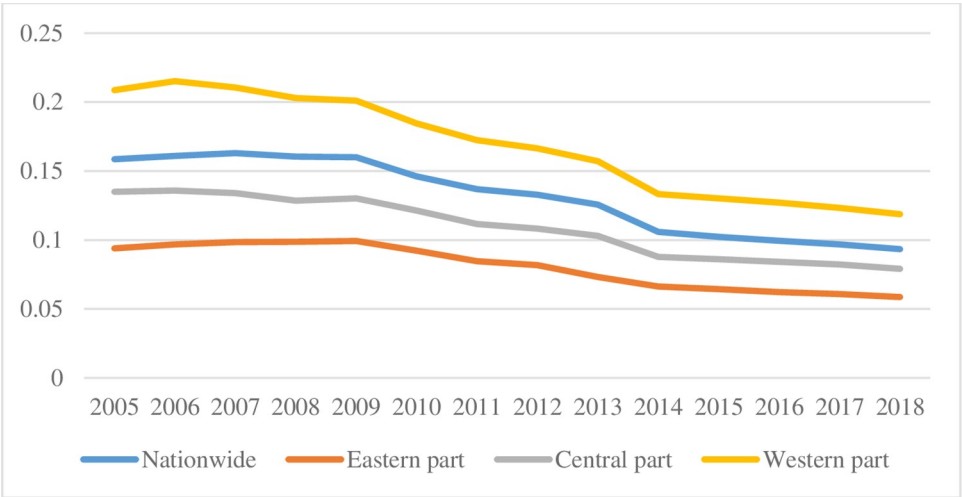

**Fig 2. Urban-Rural Thiel index by region.**

## Empirical results and analysis

### Granger's non-causality test

Mutual causality between variables may lead to endogeneity and make the estimation result biased in the regression process. Therefore, it is necessary to judge whether the urban-rural income gap has predictive effect on carbon emission reduction and pollution control by Granger non-causality test. From the results in Table 2, the urban-rural income gap is the Granger cause of carbon emission reduction and pollution control, while carbon emission reduction and pollution control is not the Granger cause of the urban-rural income gap. It indicates that the urban-rural income gap has a unidirectional causality on carbon emission reduction and pollution control.

### Baseline regression results

Tables 3 and 4 respectively list the regression results of the test models (Eqs (1) and (2)) for the impact of urban-rural income gap on carbon emission reduction and pollution control. Columns (1) and (2) show the estimation results of the random effects model (RE). Columns (3) and (4) show the estimation results of the fixed effects model (FE). From F-test results, it can be seen that the individual effect is significant, while the fixed effect is better. The result of Hausman indicates that unobservable individual effects are associated with the explanatory

**Table 1. Description of control variables.**

| Name | Variable | Definition | Data Sources |
|---|---|---|---|
| Economic growth | *GDP* | Using per capita GDP and deflating at constant 2005 prices. | A |
| Industrial structure optimization | *ISO* | Ratio of added value of tertiary sector to regional GDP. | A |
| Urbanization rate | *UR* | The proportion of urban popution. | C&D |
| Environmental regulation intensity | *ERI* | Ratio of completed investment in industrial pollution control to industrial value added. | D |
| Energy efficiency | *EE* | The energy consumption per unit of GDP. | B&D |

Note: Where A denotes China Statistical Yearbook (2007–2018), B denotes China Energy Statistical Yearbook (2007–2018), C denotes China Population and Employment Statistical Yearbook (2007–2018), and D denotes individual provincial statistical yearbooks (2007–2018).

**Table 2. Granger non-causality test results.**

| Null hypothesis | Obs | F statistics | P value |
|---|---|---|---|
| *Theil* does not Granger Cause *EQ* | 240 | 10.6044 | 0.0000 |
| *EQ* does not Granger Cause *Theil* | 240 | 1.7493 | 0.1400 |
| *GAP* does not Granger Cause *EQ* | 240 | 11.4771 | 0.0000 |
| *EQ* does not Granger Cause *Gap* | 240 | 2.3761 | 0.0528 |
| *Theil* does not Granger Cause $PerCO_2$ | 240 | 6.9273 | 0.0000 |
| $PerCO_2$ does not Granger Cause *Theil* | 240 | 0.9439 | 0.4393 |
| *GAP* does not Granger Cause $PerCO_2$ | 240 | 5.4791 | 0.0003 |
| $PerCO_2$ does not Granger Cause *GAP* | 240 | 1.5976 | 0.1758 |

variables. The random effects model (RE) regression result is biased but fixed effects model (FE) satisfies the consistency. Therefore, the FE model is chosen. Since panel data has the dual characteristics of cross-sectional and time series, the error terms may have autocorrelation, heteroskedasticity, and cross-sectional correlation. Firstly, the question of whether there is cross-sectional correlation is tested. Pearson test significantly rejects the original hypothesis that there is no contemporaneous correlation between groups and concludes that there is cross-sectional correlation in the panel data. Secondly, Modified Wald test results strongly reject the original hypothesis that there is between-group heteroskedasticity. Finally, Wool-dridge test results significantly reject the absence of intra-group autocorrelation. It is considered that there is an intra-group autocorrelation. To ensure the robustness of the regression results, we use ordinary least squares (OLS) based on Panel-Correvted Standard Error (PCSE) to estimate the test model. The results are shown in columns (5) and (6) in Tables 3 and 4.

From the regression results in Tables 3 and 4, the urban-rural income gap is positively correlated with carbon emissions per capita at least at the 10% significance level and negatively correlated with pollution control capacity at the 1% significance level. It indicates that the narrowing of the urban-rural income gap has a positive impact on carbon emission reduction at the source of production and also strengthens the pollution control capacity at the end of output. The smaller the urban-rural income gap, the better the effect of carbon emission reduction and pollution control.

**Table 3. Regressions of the urban-rural income gap on carbon emission reduction.**

| variables | RE | | FE | | PCSE | |
|---|---|---|---|---|---|---|
| | (1) | (2) | (3) | (4) | (5) | (6) |
| *Theil* | -0.0034 (0.03) | | -0.0359 (-0.30) | | 0.1797* (1.65) | |
| *GAP* | | 0.0512 (0.23) | | -0.0842 (-0.37) | | 0.4771** (2.40) |
| *GDP* | 4.1281*** (3.99) | 4.1595*** (4.32) | 2.9569*** (2.76) | 2.8719*** (2.79) | 5.9930*** (4.56) | 7.2019*** (5.89) |
| $GDP^2$ | -0.1829*** (-3.73) | -0.1842*** (-4.08) | -0.1435*** (-2.88) | -0.1394*** (-2.94) | -0.2761*** (-4.32) | -0.3330*** (-5.61) |
| *ISO* | -0.2641 (-1.58) | -0.2610 (-1.56) | -0.2228 (-1.37) | -0.2245 (-1.38) | -0.4818*** (-2.84) | -0.5496*** (-3.10) |
| *UR* | -0.0490 (-0.13) | -0.0374 (0.11) | -0.0227 (-0.05) | -0.0204 (-0.05) | 0.7938** (2.34) | 0.6269** (2.41) |
| *ERI* | 0.0601** (2.52) | 0.0604** (2.53) | 0.0310 (1.42) | 0.0306 (1.40) | 0.3856*** (9.32) | 0.3822*** (9.19) |
| *EE* | 0.3045*** (3.84) | 0.3046*** (3.83) | 0.9627*** (-7.12) | 0.9634*** (7.15) | 0.1337*** (3.39) | 0.1370*** (3.54) |
| constant | -24.1087*** (-5.39) | -24.4031*** (-5.58) | -22.414*** (-5.04) | -21.8303*** (-4.97) | -36.0531*** (-5.67) | -42.7702*** ((-6.80) |
| F | | | 45.05*** | 44.65*** | | |
| Hausman | | | 76.24*** | 76.84*** | | |
| Modified-Wald | | | | | 1.2e+05*** | 1.3e+05*** |
| Wooldridge | | | | | 0.27 | 0.27 |
| Pesaran | | | | | 1.70*** | 1.76*** |

**Table 4. Regressions of the urban-rural income gap on pollution control.**

| variables | RE | | FE | | PCSE | |
|---|---|---|---|---|---|---|
| | (1) | (2) | (3) | (4) | (5) | (6) |
| THEIL | -0.0028 (-0.21) | | 0.0012 (0.09) | | -0.0445*** (-4.31) | |
| GAP | | -0.0268 (-1.06) | | -0.0076 (-0.28) | | -0.0831*** (-3.93) |
| GDP | 3.8571*** (2.76) | 3.8414*** (2.77) | 3.5738** (2.61) | 3.5926*** (2.64) | 5.7602*** (3.18) | 6.1162*** (3.35) |
| $GDP^2$ | -0.3802*** (-2.80) | -0.3796*** (-2.81) | -0.3649*** (-2.74) | -0.3666*** (-2.76) | -0.5416*** (-3.05) | -0.5894*** (-3.30) |
| $GDP^3$ | 0.0125*** (2.85) | 0.0125*** (2.87) | 0.0124*** (2.86) | 0.0124*** (2.88) | 0.0170*** (2.94) | 0.0189*** (3.26) |
| ISO | 0.0989*** (5.31) | 0.0970*** (5.20) | 0.0884*** (4.59) | 0.0876*** (4.55) | 0.0945*** (5.73) | 0.1084*** (6.43) |
| UR | 0.0592 (1.34) | 0.0546 (1.34) | 0.1466*** (2.95) | 0.1419*** (2.92) | -0.0999** (-2.48) | -0.0427 (-1.30) |
| ERI | -0.0064** (-2.47) | -0.0064** (-2.48) | -0.0057** (-2.21) | -0.0057** (-2.21) | -0.0065* (-1.84) | -0.0069** (-1.91) |
| EE | -0.0901*** (-8.94) | -0.0909*** (-8.93) | -0.0648*** (-4.06) | -0.0653*** (-4.11) | -0.1131*** (-18.75) | -0.1129*** (-17.55) |
| Constant | -11.8421** (-2.49) | -11.6917** (-2.48) | -10.9380*** (-2.34) | -10.9758** (-2.36) | -18.4431*** (-3.00) | -19.1959*** (-3.08) |
| F | | | 34.78*** | 35.00*** | | |
| Hausman | | | 18.68*** | 17.05*** | | |
| Modified-Wald | | | | | 5276.51*** | 5241.16*** |
| Wooldridge | | | | | 48.55*** | 48.77*** |
| Pesaran | | | | | 16.23*** | 16.69*** |

Note: Z-statistics for each coefficient are in parentheses

*, **, and *** indicate 10%, 5%, and 1% significance levels, respectively.

Data source: authors' calculations.

Environmental quality not only depends on the level of income, but also on the rationality of income distribution. From the perspective of consumption, rural residents have low income levels. They are price-sensitive consumers and fail to form strong constraints on green products [31]. The improvement of rural residents' income is used to consume products with low prices but high energy consumption and pollution. Moreover, the marginal propensity to consume is higher in rural areas. Therefore the increase in income level of rural residents will substantially increase pollution-based consumption. Urban residents have high income and education levels. The preference is strong for green products, so urban residents will increase the consumption level of clean products. However, because of the small marginal propensity to consume of urban residents, the improvement of urban residents' income has limited effect on the consumption of cleaning products. In summary, the increase in clean consumption by urban residents is smaller than the increase in pollution consumption by rural residents for the same intensity of income level increase. The widening urban-rural income gap will further increase the marginal propensity to consume of urban and rural residents. The consumption demand for green products is low compared to the consumption of polluting products. There is insufficient incentive that enterprises switch to green production. Therefore, it is not conducive to enterprises to choose clean energy and improve pollution control technology. From the investment perspective, the widening urban and rural income gap makes urban residents have a higher marginal savings rate and investment. Thereby, it increases the capital stock of enterprises and further stimulates them to engage in technological innovation. In fact, technological innovation is biased in the production process. It can be divided into production-based innovation and emission reduction and pollution control innovation. The former emphasizes productivity gains and increased profits, while the latter emphasizes the protection of the ecological environment. The results of this paper suggest that the increased investment by urban residents is mostly used to promote profit-oriented production rather than environmentally friendly production by firms. At the same time, with the optimization of urban

industrial structure and the strengthening of environmental supervision, enterprises may use the amount of investment by residents for government rent-seeking or pollution transfer to reduce the cost of emission reduction and pollution control in order to avoid environmental control. That lead to some high energy consumption and high pollution enterprises being moved to rural areas. Not only does it greatly damage the rural environment, but it also seriously hinders the overall process of reducing pollution. Therefore, narrowing the income gap is conducive to promoting the use of clean energy at the source of production and the improvement of pollution control technology at the end of production. Thus promoting the reduction of emissions and pollution control, which is beneficial to the improvement of environmental quality.

(1) Economic growth (*GDP*). Economic growth has a significant inverted "U" curve relationship with per capita carbon emissions, while it has a positive "N" curve relationship with end pollution control at the 1% significance level. This shows that economic growth can achieve a win-win situation with carbon emission reduction and pollution control. With the economic development and industrial structure transformation, China's industrial center of gravity has gradually realized the transfer from the primary industry to the secondary industry and then to the tertiary industry. In the early stage of industrialization, the primary industry accounted for a large proportion, and industrialization developed steadily but on a small scale. The consumption of fossil fuels and the discharge of production wastes were relatively small. So economic growth not cause greater pressure on the ecological environment and natural resources. Along with the advance of industrialization, the second industry gradually dominate. To achieve rapid economic growth, "extravagance" pattern of industrial development, especially the heavy chemical industry rapid development, such as dependence on resources and environmental damage is stronger. So the phase of economic growth at the expense of the environment quality. With the further deepening of industrialization, China's industrial structure has realized diversified development. The improvement of technology and innovation level continue to promote the transformation of traditional industries and the vigorous development of high-tech industries. At the same time, our society is gradually developing towards digitalization, intelligence and ecology. The society finally realizes the mutual benefit of environmental quality and economic growth.

(2) Industrial structure optimization (*ISO*). The optimization of industrial structure has a significant negative correlation with per capita carbon emissions and a significant positive correlation with pollution control capacity. It indicates that the upgrading of industrial structure has a significant promoting effect on carbon emission reduction and pollution control. The development of tertiary industry can effectively improve the status quo of ecological environmental pollution. On the one hand, the characteristics of low energy consumption, emissions and pollution of the tertiary industry can greatly alleviate the contradiction among the current economic growth, environmental pollution and resource constraints. On the other hand, the development of information technology service industry also can improve the energy utilization and pollution control efficiency of polluting enterprises, further promote the transformation and upgrad the traditional industries. Therefore, the development of the tertiary industry can achieve the dual effect of carbon emission reduction and pollution control.

(3) Urbanization rate (*UR*). When the urban and rural income gap is measured by Thiel index, urbanization has a significant negative impact on carbon emission reduction at source and increasing pollution control capacity. However, when the urban-rural income ratio is used as the measurement index of the urban-rural income gap, the urbanization process has a significant positive correlation with the per capita carbon emissions, but has an insignificant negative correlation with the end governance intensity. It shows that urbanization development is not conducive to the promotion of carbon emission reduction and pollution control.

The advancement of urbanization may affect the ecological environment through scale and agglomeration effect. From the perspective of scale effect, with the influx of surplus rural labor, the population in urban areas rapidly expands, generating a large amount of demand for housing, transportation and other resources, resulting in traffic congestion, resource waste, environmental pollution and other "urban diseases". From the perspective of agglomeration effect, industrial and economic agglomeration caused by population agglomeration can realize resource sharing and environmental co-governance. Therefore, improving resource utilization efficiency and environmental governance efficiency, which is beneficial to the improvement of ecological environment quality. According to the results of this study, the impact of urbanization on environmental quality in China is still dominated by scale effect, and the positive externality of agglomeration effect has not been fully exerted. Therefore, in the process of urbanization promotion, we can encourage technological innovation, develop sharing economy and optimize the construction of public infrastructure to give full play to the positive effect of agglomeration on environmental quality.

(4) Environmental regulation intensity (*ERI*). The intensity of environmental regulation has a significant positive correlation with per capita carbon emissions, and a significant negative correlation with pollution control ability. Generally speaking, the increasing of investment in industrial pollution control means the increasing of government pollution control efforts, so as to reduce environmental pollution. However, the empirical results show that the increase of investment in environmental control is not conducive to the promotion of carbon emission reduction and pollution control. The government's efforts are increasing to regulate the overall industrial emissions of the society. However, due to the lack of precision and pertinence of the investment in environmental protection, the energy utilization rate and pollution treatment rate of enterprises are still not high. It results the mismatch between the government's investment in environmental protection and the purpose. Therefore, the government should improve the investment mechanism of environmental governance and improve the utilization rate of industrial pollution governance investment.

(5) Energy efficiency (*EE*). Energy consumption per unit GDP has a significant positive impact on per capita carbon emissions, and a significant negative impact on terminal pollution control. This indicates that the higher the energy consumption, the lower the energy efficiency and the greater the negative impact on carbon emission reduction and pollution control. Energy is one of the important inputs for economic development, but a large amount of consumption will cause waste of resources and air pollution [32]. The transformation from a major manufacturer to a strong manufacturer means that an extensive model of development characterized cannot be sustained by high energy consumption, high pollution and high emissions. Therefore, enterprises must improve energy use efficiency and reduce pollution emissions through technological progress, innovative production and economic management mode.

## Robustness test

Severe endogeneity can lead to bias in estimation results. Therefore, this problem is considered in the process of empirical analysis. It is generally believed that endogeneity may arise from the cross effects of explanatory and explained variables, omitted variables, and computational errors. According to the results of Granger non-causality test, it can be determined that there is no bidirectional causality. Therefore, we focus on the effects of omitted variables and computational errors on the estimation results. From the omitted variables, we could not include the variables that so affect the emission reduction and pollution control in the regression equation. From the calculation error, as only eight basic energy sources were used in the

**Table 5. Instrumental variable method regression results.**

| variables | $PerCO_2$ | $PerCO_2$ | EQ | EQ |
|---|---|---|---|---|
| | (1) | (2) | (3) | (4) |
| Theil | 0.2274* (1.89) | | -0.0471*** (-4.02) | |
| GAP | | 0.5032** (2.07) | | -0.0910*** (-3.88) |
| GDP | 5.8961*** (3.71) | 7.6688*** (4.75) | 6.2574** (2.16) | 6.6604** (2.31) |
| $GDP^2$ | -0.2815*** (-3.74) | -0.3639*** (-4.78) | -0.5897** (-2.10) | -0.6444** (-2.30) |
| $GDP^3$ | | | 0.0185** (2.05) | 0.0208** (2.30) |
| ISO | -0.5716*** (-2.99) | -0.6325*** (-3.33) | 0.0898*** (4.47) | 0.1036*** (5.15) |
| UR | 1.4642*** (3.40) | 1.1464*** (3.22) | -0.0883** (-2.11) | -0.0233 (-0.67) |
| ERI | 0.4078*** (11.23) | 0.4063*** (11.22) | -0.0048*** (-1.29) | -0.0049*** (-1.31) |
| EE | 0.1710*** (3.58) | 0.1698*** (3.57) | -0.1123*** (-22.95) | -0.1118*** (-22.87) |
| Content | -37.4867*** (-4.79) | -46.8017*** (-5.61) | -20.1998** (-2.04) | -21.0391** (-2.12) |

calculation of carbon emissions. In this paper, the two-stage least squares (Panel Data-2SLS) method is chosen to test the robustness of the above results by using the explanatory variables with one lag period as its own instrumental variable. The regression results of 2SLS are shown in Table 5.

The results in Table 5 show that the urban-rural income gap is positively correlated with per capita carbon emissions, while it is negatively correlated with end-of-pipe pollution control. This suggests that after controlling for endogeneity, the narrowing of the urban-rural income gap is still conducive to the advancement of the carbon emission reduction and pollution control. The robustness of the above regression results is enhanced. For the other control variables, there are also previous results that remain consistent.

## Analysis of regional heterogeneity

The uneven development of the regions restricts the promotion of the high-quality development level of the national economy. Solving the problem of imbalance and insufficiency through the whole process of China's development. Therefore, combined with China's national conditions, we divide the country into eastern, central and western regions for heterogeneity analysis based on China Statistical Yearbook. Tables 6 and 7 respectively show the regression results of the urban-rural income gap on pollution control and carbon emission reduction in eastern, central and western region.

**Table 6. Results of the urban-rural income gap on carbon emission reductions by region.**

| variables | Eastern part | | Central part | | Western part | |
|---|---|---|---|---|---|---|
| | (1) | (2) | (3) | (4) | (5) | (6) |
| Theil | -0.7957*** (-13.35) | | 0.7893*** (4.18) | | -0.2886 (-0.93) | |
| GAP | | -1.8632*** (-10.19) | | 1.8426*** (4.68) | | -0.4004 (-0.83) |
| GDP | 7.9108*** (4.34) | 2.8613*** (-3.07) | 0.6650*** (3.39) | 0.7302*** (3.77) | -3.1974* (-1.74) | -3.8106* (-1.93) |
| $GDP^2$ | -0.3721*** (-4.52) | -0.1504* (-1.70) | | | 0.1680* (1.84) | 0.1988** (2.05) |
| ISO | -0.9536*** (-7.35) | -0.6440*** (-5.34) | -1.2977*** (-4.18) | -1.342*** (-4.36) | 0.0985 (0.35) | 0.1128 (0.40) |
| UR | -0.9749*** (-3.43) | 0.8463*** (2.94) | 2.3546*** (5.82) | 2.0753*** (5.72) | 2.0012*** (3.14) | 2.1588*** (3.63) |
| ERI | 0.0965*** (3.43) | 0.0885*** (3.05) | 0.4254*** (6.20) | 0.4097 (6.21) | 0.4470*** (5.52) | 0.4446*** (5.50) |
| EE | 0.1606*** (4.97) | 0.2350*** (6.93) | 0.1774** (2.31) | 0.1469* (1.93) | 0.1178 (1.26) | 0.1205 (1.29) |
| Content | -40.6912*** (-4.51) | -16.1929 (-1.62) | -16.480*** (-8.66) | -19.3321*** (-8.67) | 6.4626 (0.70) | 9.9252 (0.95) |
| $R^2$ | 0.7061 | 0.6685 | 0.7133 | 0.7296 | 0.5951 | 0.5946 |

**Table 7. Results of the urban-rural income gap on pollution control by region.**

| variables | Eastern part | | Central part | | Western part | |
|---|---|---|---|---|---|---|
| | (1) | (2) | (3) | (4) | (5) | (6) |
| *Theil* | -0.0318*** (-3.73) | | -0.0310 (-1.42) | | -0.0807*** (-5.63) | |
| *GAP* | | -0.1266*** (-4.88) | | -0.0732 (-1.62) | | -0.1102*** (-4.94) |
| *GDP* | 0.0626*** (4.18) | 0.0461*** (3.01) | 27.4769** (1.97) | 28.1168** (2.03) | 1.1708*** (7.74) | 1.0018*** (5.84) |
| $GDP^2$ | | | -2.8084** (-2.00) | -2.8742** (-2.06) | -0.0519*** (-7.07) | -0.0434*** (-5.24) |
| $GDP^3$ | | | 0.0954** (2.02) | 0.0976** (2.08) | | |
| *ISO* | -0.0220 (-0.97) | -0.0029 (-0.13) | 0.1001** (2.17) | 0.1010** (2.21) | 0.1079*** (5.17) | 0.1120*** (5.41) |
| *UR* | 0.0467 (1.32) | 0.1034*** (3.31) | 0.2013*** (3.11) | 0.2138*** (3.80) | -0.3569*** (-7.66) | -0.3122*** (-6.27) |
| *ERI* | 0.0003 (0.06) | -0.0017 (-0.35) | -0.0274*** (-3.58) | -0.0268*** (-3.49) | -0.0029 (-0.89) | -0.0036 (-1.11) |
| *EE* | -0.1188*** (-21.74) | -0.1165*** (-19.84) | -0.0940*** (-8.90) | -0.0923*** (-8.95) | -0.1293*** (-21.73) | -0.1284*** (-21.83) |
| Content | 0.6818*** (5.43) | 0.8259*** (6.83) | -88.5663* (-1.92) | -90.5499** (-1.97) | -3.4431*** (-4.62) | -2.4933** (-2.89) |
| $R^2$ | 0.8970 | 0.9016 | 0.7975 | 0.7988 | 0.7953 | 0.7902 |

It can be seen from Tables 6 and 7 that the impact of urban-rural income gap on pollution control in the eastern, central and western regions of China is basically consistent with the regression results of the whole country. However, the impact on per capita carbon emissions shows great regional heterogeneity. The urban-rural income gap has a significant negative correlation with per capita carbon emissions in the eastern region, a significant positive correlation in the central region, and an insignificant negative correlation in the western region. This shows that the widening of the urban and rural income gap in the eastern region is not conducive to pollution control, but has a significant promoting effect on carbon emission reduction. Based on the analysis from a national perspective, we believe that the widening urban and rural income gap will hinder the progress of reducing pollution through consumption and investment. However, the income level of urban residents in eastern is significantly ahead of the national average level. According to the "governance of the rich" theory, with the increase of income level, the environment gradually turns from a luxury to a necessity. More "rich people" begin to strengthen environmental protection [33]. Therefore, the environmental quality in eastern region is affected by both the "governance by the rich" effect and the conduction effect of consumption and investment. At the same time, due to the difference of policy bias and the difficulty of pollutant treatment, the urban-rural income gap only plays a positive role in the reduction of carbon emissions, but still has a negative effect on pollutant treatment in the eastern region. The central region has basically maintained the national average level in economic development, income, science, technology, and education. Therefore, it is consistent with the overall analysis results of the country. The widening of urban and rural income gap in western has a significant inhibiting effect on pollution treatment, but has no obvious effect on carbon emission reduction. Due to the low level of economic development and income of urban and rural residents, as well as abundant natural resources such as geothermal energy, wind energy and solar energy, the carbon emission reduction is at a relatively low level in the western region. Therefore, the change of urban-rural income gap does not have a significant impact on carbon emissions reduction. In terms of terminal pollution control, the widening of income gap between the eastern, central and western regions is not conducive to the improvement of pollution control. Although the regression results in the central region are not statistically significant, they can still reflect the important impact of the urban-rural income gap on the terminal pollution control partly.

(1) Economic growth (*GDP*). There is regional heterogeneity between economic growth and per capita carbon emissions and terminal pollution control. There is an inverted "U" curve

between economic growth and per capita carbon emissions in eastern region, and a positive linear relationship between economic growth and pollution control. This shows that under the current economic development model, economic growth can achieve a win-win situation between carbon emission reduction and pollution control in the eastern region. There is a positive linear relationship between economic growth and per capita carbon emissions, and a positive "N" type relationship between economic growth and pollution control in central region. This shows that economic growth and pollution control can achieve a win-win situation in the central region. However, it will still lead to the increase of per capita carbon emissions. The relative decoupling has not been realized between economic growth and carbon emissions [34]. There is a positive "U" curve between economic growth and per capita carbon emission, and an inverted "U" curve between economic growth and terminal pollution control in the western region. This indicates that the current economic development model cannot achieve the win-win situation of economic growth and carbon emission control or pollution control in western region. Therefore, the western region learn from the development of the eastern and central regions. At the same time, the western region can combine with its own advantages, encourage geothermal, wind, solar energy and other clean and renewable energy development.

(2) Industrial structure optimization (*ISO*). Industrial structure optimization has a significant negative impact on per capita carbon emissions in the eastern and central regions, but not in the western regions. The regression coefficient of terminal pollution treatment is not positive in the central and western regions, but not in the eastern region. This indicates that industrial structure optimization has a more obvious promoting effect on source carbon emission control in eastern region. In the western region, the effect was more obvious on promoting the end treatment intensity. However, in the central region, carbon emission control and pollution control have a significant promoting effect. Therefore, each region should choose environmental protection schemes according to its own economic basis, industrial characteristics and environmental conditions.

(3) Urbanization rate (*UR*). Urbanization process has a significant positive impact on carbon emissions per capita in eastern, central and western region. The regression coefficient is significantly positive in the eastern and central regions, but is significantly negative in the western regions. In terms of per capita carbon emissions, the scale effect of urbanization in the eastern and central regions is greater than the agglomeration effect. In terms of pollution control, the scale effect of urbanization in the eastern and central regions is smaller than the agglomeration effect. However, due to the limitations of natural conditions and industrial structure in western region, urbanization has not yet achieved efficient development. Therefore, urbanization is still focused on giving full play to the scale effect. Significant progress has not been made in carbon emission reduction and pollution control.

(4) Environmental regulation intensity (*ERI*). The intensity of environmental regulation is positively correlated with carbon emissions per capita in the eastern and central regions, but not in the western regions. For the pollution control ability, the regression coefficient is negative only in the central region, but not in the eastern and western regions. This indicates that the increase in the intensity of environmental regulation has a significant inhibiting effect on carbon emission control in the eastern region. In the middle of the region, carbon emission control and terminal pollution control have an obstacle effect. However, there is no significant effect on carbon emission reduction and pollution control in western region. It shows that the investment in environmental protection of local governments has not been fully and effectively utilized in China.

(5) Energy efficiency (*EE*). Energy efficiency has an obvious promoting effect on carbon emission reduction and pollution control in the three regions, which is consistent with the regression results of the whole country.

## Discussions and conclusion

### Discussions

With economic development, we should not only emphasize on sharing of economic development and narrow the urban and rural income gap, but also on the results of green economic. Based on the data of 30 provinces in China from 2006 to 2017, we build a static panel model, and use OLS parameter estimation method to systematically investigate the relationship between the urban-rural income gap and carbon emission reduction and pollution control from the national and regional levels in China. The Granger non-causal test and instrumental variable method are used to examine the two-way causality between urban-rural income gap and carbon emission reduction and pollution control, so as to ensure the robustness of the results. The results found that the narrowing of the urban-rural income gap has a significant inhibiting effect on both carbon emission reduction and pollution control across the country. At the regional level, the narrowing of the urban-rural income gap in all regions has a dampening effect on pollution control, while there is regional heterogeneity in the impact on carbon emission reduction. The results in the central and western regions are consistent with those in the whole country, but in the eastern region, widening the urban-rural income gap is beneficial to the reduction of carbon emissions due to the "rich people". The above research conclusions are of great significance to accelerate the narrowing of China's urban-rural income gap and the pace of ecological civilization construction.

### Policy implications

From the above results, it can be seen that narrowing the urban-rural income gap is not only the way towards a new type of urbanization, but also the key to reducing environmental pollution. With the rapid development of the economy, people have put forward higher requirements for a better life. While expanding the size of the "cake", the "cake" should also be reasonably divided. Under the leadership of the Party and the State, all regions should collaborate with each other and give full play to the role of radiation between regions. At the same time, each region makes development programs combined with their own development status and the advantages of resources and environment. In addition, we should promote new urbanization and new rural construction, improve education and income levels in rural areas, and thus raise the overall environmental awareness of the society. Meanwhile, Enterprises should strengthen technological innovation, realize the green transformation of production, improve the efficiency of resource utilization and environmental protection, optimize the energy use structure, and give full play to the effect of scale economy. Through resource technology sharing and complementarity, enterprises can reduce resource consumption and enterprise operating costs, promote the development and popularization of clean technologies, and then relieve the pressure on resources and environment. Individuals should also actively respond to the call for a green life. While pursuing a comfortable life, they should internalize the concept of green into their hearts and practices. In the joint efforts of the government, enterprises and individuals to achieve urban and rural integration and ecological environment construction of win-win.

### Limitations and future research

We, of course, can acknowledge the deficiencies of our study. First, when analyzing heterogeneity, we only analyze different economic development regions, such as the eastern, central and western regions. In view of the differences between urban and rural populations, different population divisions may have heterogeneous effects. Later research can be divided into more

detailed and more general conclusions can be obtained. Second, our data is based on China and may not be suitable for other countries. If the relevant data can be obtained in the later period, the later research can solve this problem.

## Supporting information

**S1 Data.**
(XLSX)

## Author Contributions

**Data curation:** Lujing Wang.

**Formal analysis:** Ming Zhang.

**Methodology:** Lujing Wang.

**Writing – original draft:** Ming Zhang.

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
