## [Decision Letter · Decision Letter 0]

9 Aug 2021

PONE-D-21-22293

Exploring the Impact of Narrowing Urban-Rural Income Gap on Carbon Emission Reduction and Pollution Control

PLOS ONE

Dear Dr. Zhang,

Thank you for submitting your manuscript to PLOS ONE. After careful consideration, we feel that it has merit but does not fully meet PLOS ONE’s publication criteria as it currently stands. Therefore, we invite you to submit a revised version of the manuscript that addresses the points raised during the review process.

We look forward to receiving your revised manuscript.

Kind regards,

Bing Xue, Ph.D.

Academic Editor

PLOS ONE

Journal Requirements:

3. PLOS requires an ORCID iD for the corresponding author in Editorial Manager on papers submitted after December 6th, 2016. Please ensure that you have an ORCID iD and that it is validated in Editorial Manager. To do this, go to ‘Update my Information’ (in the upper left-hand corner of the main menu), and click on the Fetch/Validate link next to the ORCID field. This will take you to the ORCID site and allow you to create a new iD or authenticate a pre-existing iD in Editorial Manager. Please see the following video for instructions on linking an ORCID iD to your Editorial Manager account: https://www.youtube.com/watch?v=_xcclfuvtxQ.

Reviewers' comments:

Reviewer's Responses to Questions

**Comments to the Author**

1. Is the manuscript technically sound, and do the data support the conclusions?

Reviewer #1: Yes

Reviewer #2: Yes

2. Has the statistical analysis been performed appropriately and rigorously? 

Reviewer #1: Yes

Reviewer #2: Yes

3. Have the authors made all data underlying the findings in their manuscript fully available?

Reviewer #1: Yes

Reviewer #2: Yes

4. Is the manuscript presented in an intelligible fashion and written in standard English?

Reviewer #1: Yes

Reviewer #2: Yes

5. Review Comments to the Author

Reviewer #1: This paper is very interesting and value. It is recommneded for publication after minor revision.

1: Can the innovation of the paper be summarized into three points?

2: The abstract can add to the research background of this article and raise the question.

3: The paper can add some discussions.

4: The research deficiencies of this article can be put forward.

Reviewer #2: Based on the inter-provincial panel data from 2006-2017, the paper empirically analyzes the impact of urban-rural income gap on emission reduction and pollution control, and some enlightening conclusions are obtained. The topic is very important and is one to which the authors have made a significant contribution. Thus, I have no hesitation in recommending that it be accepted for publication after s few typos and other minor errors and details have been attend to. My specific comments are presented as follows:

1. I would strongly advise the authors to pay attention to linguistic alterations. The paper would benefit from some closer proofreading. It includes many linguistic errors (e.g. agreement verbs) that at times make it difficult to follow. This should be addressed throughout the manuscript.

2. There are some references with improper punctuation, and it is recommended that they be carefully proofread.

3. The full name of abbreviations must be presented in the abstract.

4. In the introduction section, the authors provide a comprehensive literature review on the current research, but what is the main contribution of this paper? It's not clear in the current version. Similarity and differences between this study and the existing literature should be discussed and some main contribution of this paper should be clearly highlighted.

5. The table format is not standardized enough, and it is suggested that the table format of this paper should be revised.

6. All the carbon emission factors and oxidation rate of different fuel type should be presented in this paper.

7. I recommend to publish this paper after minor revision.

6. PLOS authors have the option to publish the peer review history of their article (what does this mean?). If published, this will include your full peer review and any attached files.

Reviewer #1: No

Reviewer #2: No

---

## [Author Response · Author response to Decision Letter 0]

30 Sep 2021

Reviewer #1: This paper is very interesting and value. It is recommneded for publication after minor revision.

1: Can the innovation of the paper be summarized into three points?

Yes, according to the reviewer’s advice, the innovations of this article are summarized into three points.

(1) In the selection of pollutants, both source emission reduction and end-of-pipe pollution control are considered to supplement and improve the existing studies. At the input side, most industries' production activities are dominated by the consumption of fossil energy and carbon emissions are higher. Therefore, carbon emissions are selected as the variable for carbon emission control at the production source. At the output side, the entropy weight method is used to construct the pollution control capacity index as the variable of pollution control at the output side based on the data of industrial three waste emissions. (2) Considering the possible two-way causality between urban-rural income gap and environmental pollution, this paper uses Granger non-causality test for ranking. And the lagged first order of urban-rural income gap is used as an instrumental variable to alleviate the endogenous problems in the model. (3) The relationship between urban-rural income disparity and emission reduction and pollution control was analyzed heterogeneously about the eastern, central and western regions in China.

2: The abstract can add to the research background of this article and raise the question.

Yes, according to the reviewer’s advice, background and questions were raised in the summary section.

Over the past four decades, China have experienced rapid economic growth but also a widending urban-rural income gap and deteriorating air quality. Based on the panel data of 30 provinces in China from 2006 to 2017, this paper investigates the effect of narrowing the urban-rural income gap on carbon emission reduction and pollution control by using OLS method. The empirical results indicate that: the narrowing of the urban-rural income gap has a positive impact on pollution control, while there are regional differences in the impact on carbon emission reduction. In the perspective of the whole country and central and western regions, the narrowing of the urban-rural income gap is conducive to carbon emission reduction. However, the narrowing of the urban-rural income gap increases carbon emissions in the eastern regions where economic development is at high level. This paper provides a theoretical basis and policy reference for promoting urban-rural integration and construction of ecological civilization.

3: The paper can add some discussions.

Yes, according to the reviewer’s advice, we have added a discussion part to the fourth part of the article. For details, please see page 20 of the article.

4: The research deficiencies of this article can be put forward.

Yes, according to the reviewer’s advice, We present the deficiencies of this study.

We, of course, can acknowledge the deficiencies of our study. First, when analyzing heterogeneity, we only analyze different economic development regions, such as the eastern, central and western regions. In view of the differences between urban and rural populations, different population divisions may have heterogeneous effects. Later research can be divided into more detailed and more general conclusions can be obtained. Second, our data is based on China and may not be suitable for other countries. If the relevant data can be obtained in the later period, the later research can solve this problem.

Reviewer #2: Based on the inter-provincial panel data from 2006-2017, the paper empirically analyzes the impact of urban-rural income gap on emission reduction and pollution control, and some enlightening conclusions are obtained. The topic is very important and is one to which the authors have made a significant contribution. Thus, I have no hesitation in recommending that it be accepted for publication after s few typos and other minor errors and details have been attend to. My specific comments are presented as follows:

1. I would strongly advise the authors to pay attention to linguistic alterations. The paper would benefit from some closer proofreading. It includes many linguistic errors (e.g. agreement verbs) that at times make it difficult to follow. This should be addressed throughout the manuscript.

We have carefully proof-readed the manuscript again to minimize typographical, grammatical, and bibliographical errors. In addition, your comments are valuable and very helpful for revising and improving our paper, as well as the important guiding significance to our researches. We have studied comments carefully and have made correction which we hope meet with approval.

2. There are some references with improper punctuation, and it is recommended that they be carefully proofread.

We did it through an oversight. The form of Table 1 has been modified for better reading in the revised manuscript. Please refer to Table 3 of the revised manuscript.

3. The full name of abbreviations must be presented in the abstract.

Yes, according to the reviewer’s advice, the full name of acronyms have added in this paper.

4. In the introduction section, the authors provide a comprehensive literature review on the current research, but what is the main contribution of this paper? It's not clear in the current version. Similarity and differences between this study and the existing literature should be discussed and some main contribution of this paper should be clearly highlighted.

 Yes, according to the reviewer’s advice, the main contribution of this paper has added in Introduction, as shown in the second paragraph on page 5.

5. The table format is not standardized enough, and it is suggested that the table format of this paper should be revised.

We did it through an oversight. The form of table has been modified for better reading in the revised manuscript. Please refer to the tables of the revised manuscript.

6. All the carbon emission factors and oxidation rate of different fuel type should be presented in this paper.

The authors’ answer: The carbon emission factors and oxidation rate of different fuel type are collected from IPCC (2006). Please refer to IPCC (2006).

7. I recommend to publish this paper after minor revision.

The authors’ answer: Thank you for your recognition of our paper. Your comments are valuable and very helpful for revising and improving our paper, as well as the important guiding significance to our researches.

---

## [Editor Report · Decision Letter 1]

19 Oct 2021

Exploring the Impact of Narrowing Urban-Rural Income Gap on Carbon Emission Reduction and Pollution Control

PONE-D-21-22293R1

Dear Dr. Zhang,

We’re pleased to inform you that your manuscript has been judged scientifically suitable for publication and will be formally accepted for publication once it meets all outstanding technical requirements.

Kind regards,

Bing Xue, Ph.D.

Academic Editor

PLOS ONE